# Adaptation to CT Reconstruction Kernels by Enforcing Cross-Domain Feature Maps Consistency

**DOI:** 10.3390/jimaging8090234

**Published:** 2022-08-30

**Authors:** Stanislav Shimovolos, Andrey Shushko, Mikhail Belyaev, Boris Shirokikh

**Affiliations:** 1Moscow Institute of Physics and Technology, 141701 Moscow, Russia; 2Skolkovo Institute of Science and Technology, 143026 Moscow, Russia; 3Artificial Intelligence Research Institute (AIRI), 105064 Moscow, Russia

**Keywords:** chest computed tomography, convolutional neural network, COVID-19 segmentation, domain adaptation

## Abstract

Deep learning methods provide significant assistance in analyzing coronavirus disease (COVID-19) in chest computed tomography (CT) images, including identification, severity assessment, and segmentation. Although the earlier developed methods address the lack of data and specific annotations, the current goal is to build a robust algorithm for clinical use, having a larger pool of available data. With the larger datasets, the domain shift problem arises, affecting the performance of methods on the unseen data. One of the critical sources of domain shift in CT images is the difference in reconstruction kernels used to generate images from the raw data (sinograms). In this paper, we show a decrease in the COVID-19 segmentation quality of the model trained on the smooth and tested on the sharp reconstruction kernels. Furthermore, we compare several domain adaptation approaches to tackle the problem, such as task-specific augmentation and unsupervised adversarial learning. Finally, we propose the unsupervised adaptation method, called F-Consistency, that outperforms the previous approaches. Our method exploits a set of unlabeled CT image pairs which differ only in reconstruction kernels within every pair. It enforces the similarity of the network’s hidden representations (feature maps) by minimizing the mean squared error (MSE) between paired feature maps. We show our method achieving a 0.64 Dice Score on the test dataset with unseen sharp kernels, compared to the 0.56 Dice Score of the baseline model. Moreover, F-Consistency scores 0.80 Dice Score between predictions on the paired images, which almost doubles the baseline score of 0.46 and surpasses the other methods. We also show F-Consistency to better generalize on the unseen kernels and without the presence of the COVID-19 lesions than the other methods trained on unlabeled data.

## 1. Introduction

After the coronavirus disease (COVID-19) outbreak, a broad spectrum of automated algorithms have been developed to assist in the clinical analysis of the virus [1]. Among others, we consider analyzing chest computer tomography (CT) images. Firstly, CT imaging de facto has become one of the reliable clinical pretests for COVID-19 diagnosis [2]. Secondly, well-developed deep learning techniques for volumetric CT processing allow precise and efficient analysis of the different COVID-19 markers. The latter includes identification [3], prognosis [4], severity assessment [5], and detection or segmentation of the consolidation or ground-glass opacity.

One of the easiest to interpret and clinically useful markers is segmentation [6]. Segmentation provides us with classification, severity estimation [7], or differentiating from other pathologies in a straightforward manner by evaluating the output mask. However, training a segmentation model takes huge efforts in terms of voxel-wise annotations. The earlier developed models have faced the lack of publicly available data annotated with segmentation masks. To achieve the high segmentation quality, more sophisticated methods have been designed, e.g., solving a multitask problem and merging datasets with different annotations [8]. Now, a larger pool of COVID-19 segmentation datasets is available, e.g., [9]; therefore, the current goal is to build a robust algorithm for clinical use.

In merging a larger pool of data, the problem of *domain shift* arises. Domain shift is one of the most salient problems in medical computer vision [10]. A model trained on the data from one distribution might yield poor results on the data from a different distribution. In CT imaging, one of the main sources of the domain shift is the difference in *reconstruction kernels*. Here, the reconstruction kernel is a parameter of the Filtered Back Projection (FBP) reconstruction algorithm [11]. One could perceptually compare the same image reconstructed with two different kernels in Figure 3, e.g., B1 and B2. For the kernel-caused domain shift, several works have shown the deterioration of the quality of the model in lung cancer segmentation [12] and in emphysema segmentation [13].

In this paper, we show that the domain shift induced by the difference in reconstruction kernels decreases the quality of the COVID-19 segmentation algorithms. To do so, we construct two domains from the publicly available data: the *source* domain with the *smooth* reconstruction kernels and *target* domain with the *sharp* reconstruction kernels. We train the segmentation model on the source domain and test it on the target domain. We then validate the most relevant domain adaptation methods with the observed decrease in test scores. In our comparison, we include an augmentation approach [14], unsupervised adversarial learning [15], and also our proposed method that couples with the considered domain shift problem more efficiently.

All methods except the augmentation one require additional unlabeled data from the *target* domain. In this task, a large pool of unlabeled chest CT image pairs which differ only in reconstruction kernels within every pair is publicly available, e.g., [16]. The intuition here is that the adaptation methods should outperform the augmentation one when a broader range of real-world data is available. Specifically, in such conditions, we propose enforcing the cross-domain feature maps consistency between paired images; we call our method *F-Consistency* (Figure 1). F-Consistency minimizes the mean squared error (MSE) between the network’s hidden representations (feature maps) of paired images. We expect that explicitly enforcing consistency on the paired images should outperform the adversarial learning that emulates similar behavior minimizing the adversarial loss.

We also note that our method could be scaled on the other tasks, such as classification, detection, or multitasking, without any restrictions. Below, we discuss the most relevant works to our method, then summarize our contributions.

### 1.1. Related Work

We begin with discussing the task-specific augmentation approaches since they are a straightforward solution to the domain shift problem. Contrary to the classical augmentation techniques for CT images like windowing [17] or filtering [18], the authors of [14] the proposed method called FBPAug. Notably, the latter augmentation technique directly approximates our domain shift. The authors also showed that FBPAug outperforms other augmentations. Thus, we consider FBPAug as one of the solutions and compare it with the other methods.

With the unlabeled target data, we can apply unsupervised domain adaptation methods to improve the model’s performance on the target domain. The adversarial approaches are shown to outperform other methods [15]. Considering the paired nature of our data, we divide the adversarial methods into two groups: (i) image-to-image translation and (ii) feature-level adaptation.

The first group of methods aims to translate an image from source to target domain. In [19], the authors used CycleGAN for unpaired image-to-image translation. Further, this method was implemented both for the MRI [20] and CT [21] image translations. The paired image-to-image translation is closer to our setup. Such a method requires image pairs that have a different style but the same semantic content. CT images that differ only in the reconstruction kernel correspond to this setup. Here, the authors of [13] and [12] proposed a convolutional neural network (CNN) to translate the images reconstructed with one kernel to another kernel. However, we need to train a separate model for every pair of kernels. Our setup includes seven known kernels that already yield training of 42 translation models. The number of models grows in quadratic progression depending on the number of kernels. Therefore, the image translation methods lack the generalization ability in our setup and we do not consider them.

The feature-level adaptation methods are independent of the number of domains. Mostly, these methods are based on adversarial learning as in [15]. The latter approach also finds several successful applications in medical imaging, e.g., [22,23]. As these methods are conceptually close to each other, we stick with implementing a domain adversarial neural network (DANN) from [15].

The idea of F-Consistency is conceptually close to self-supervised learning, where the unlabeled data is used to train a model using pseudo-labels. In [24], the authors described different pretext tasks in medical imaging. Similar to our approach, the authors [25] enforce the model consistency for the initial and augmented images at the prediction and feature levels. Furthermore, the authors [26] extended the self-supervised methodology to solve a domain adaptation problem. However, the goal of the self-supervised methods is to use a large collection of images without annotations to improve the model’s performance on the source domain. Contrary to self-supervised learning, our goal is to achieve the highest possible performance on the target domain.

Finally, we note that there is no standardized benchmark for the COVID-19 segmentation task [27]. However, we also note that most of the COVID-19 segmentation approaches use the same U-Net-based model; thus, our method could be generalized to all these setups. Moreover, our domain adaptation method could be translated from segmentation to classification or detection tasks.

### 1.2. Contributions

Our work highlights a domain shift problem in the COVID-19 segmentation task and suggests an efficient solution to this problem. These are our three main contributions:Firstly, we demonstrate that the difference in CT reconstruction kernels affects the segmentation quality of COVID-19 lesions. The model without adaptation achieves only a 0.56 Dice Score on the unseen domain, while the best adaptation methods scores 0.64. In terms of similarity between predictions on the paired images, the baseline Dice Score is 0.46, which is almost two times lower than the 0.80 achieved by our method.Secondly, we adopt a series of adaptation approaches to solve the highlighted problem and extensively compare their performance under the different conditions.Thirdly, we propose the flexible adaptation approach that outperforms the other considered methods when provided with enough unlabeled data. We also show that our method better generalizes to unseen CT reconstruction kernels and it is less sensitive to the absence of the semantic content (COVID-19 lesions) in the unlabeled data than the other methods trained on unlabeled data.

## 2. Method

In this paper, we consider solving a binary segmentation task, where the positive class is the voxels of volumetric chest CT image with consolidation or ground-glass opacity. All methods are built upon the convolutional neural network, which we detail in Section 2.1.

We train these methods using the annotated dataset $S_{s}={\left\{\left(x_{i},y_{i}\right)\right\}}_{i=1}^{N_{s}}$, where *x* is a volumetric CT image, *y* is a corresponding binary mask, and $N_{s}$ is the total size of training dataset. The dataset $S_{s}$ consists of images reconstructed with smooth kernels; we call it source dataset. We test all methods using the annotated dataset $S_{t}={\left\{\left(x_{i},y_{i}\right)\right\}}_{i=1}^{N_{t}}$. The dataset $S_{t}$ consists of the $N_{t}$ images reconstructed with the sharp kernels; we call it target dataset. Although $S_{t}$ contains annotations, we use them only to calculate the final score.

In Section 2.2, we describe the only adaptation method, FBPAug, that uses no data except the source dataset. The other methods use additional paired dataset $S_{2}={\left\{\left(x_{i},{\tilde{x}}_{i}\right)\right\}}_{i=1}^{N_{2}}$, which has no annotations. However, every image $x\in S_{2}$ has a paired image $\tilde{x}$ reconstructed from the same sinogram but with a different kernel. Here, we assume that *x* belongs to the source domain and $\tilde{x}$ belongs to the target domain. Now, the problem can be formulated as unsupervised domain adaptation, and we detail the corresponding adversarial training approach in Section 2.3. We also propose enforcing the similarity between the feature maps of paired images (Section 2.4). In Section 2.5, we detail enforcing the similarity between predictions.

### 2.1. COVID-19 Segmentation

In all COVID-19 segmentation experiments, we use the same 2D U-Net architecture [28] trained on the axial slices. We do not use a 3D model for two reasons. Firstly, as we show in Section 3.1, the images have a large difference in the inter-slice distances (from 0.6 to 8.0 mm), which can affect the performance of the 3D model. Secondly, the authors of [8] have shown 2D and 3D models yielding similar results in the same setup with various inter-slice distances. Moreover, we note that all considered methods are independent of the architecture choice. We also introduce the standard architectural modifications, replacing every convolution layer with the Residual Block [29]. To train the segmentation model, we use binary cross-entropy (BCE) loss.

### 2.2. Filtered Backprojection Augmentation

The first adaptation method that we consider is a task-specific augmentation called FBPAug [14]. FBPAug emulates the CT reconstruction process with different kernels; thus, it might be a straightforward solution to the considered domain shift problem.

However, FBPAug gives us only an approximate solution, which is also restricted by the choice of kernel parameterization. We describe FBPAug as a three-step procedure for a given image $x\in S_{s}$. Firstly, it applies a discrete Radon transform to the image. Secondly, it convolves the transformed image with the reconstruction kernel. The kernel is randomly sampled from the predefined parametric family of kernels on every iteration. Thirdly, it applies the back-projection operation to the result and outputs the augmented image FBP(x). A complete description of the method can be found in [14].

We outline three weak spots in the FBPAug pipeline that motivate us to use the other domain adaptation approaches. As described above, FBPAug applies two discrete approximations, a discrete Radon transform and back-projection, leading to the information loss. Furthermore, the original convolution kernels used by CT manufacturers are unavailable, and the parametric family of kernels proposed in [14] is also an approximation. Finally, modern CT systems typically employ model-based iterative reconstruction methods instead of just raw FBP thus, FBPAug covers the reconstruction process only partially.

In the context of described weaknesses, we expect FBPAug to perform worse than the other adaptation methods when a wide range of unlabeled data is available for the latter. Nonetheless, FBPAug improves the consistency scores in [14], and we consider it one of the main adaptation approaches.

### 2.3. Domain Adversarial Neural Network

Further, we detail the methods that work with the unlabeled pool of data $S_{2}$. As mentioned at the beginning of the section, the problem can be reformulated as unsupervised domain adaptation. The most successful approaches to this problem are based on adversarial training. Therefore, we adopt the approach of [15] and build a *domain adversarial neural network* (DANN).

DANN includes an additional domain classifier or discriminator which aims to classify images between the source and target domains using their feature maps. We train the model to minimize the loss on the primary task (segmentation) and simultaneously maximize the discriminator’s loss. Thus, the segmentation part of the model learns domain features that are indistinguishable from the discriminator. The latter should improve the performance of the model on the domain.

The original DANN architecture consists of three parts: (i) feature extractor $H_{f}$, the part of segmentation model that maps input images *x* into the feature space; (ii) segmentation head $H_{p}$, the complement part of segmentation model that predicts binary mask $\hat{y}=H_{p}\left(H_{f}\left(x\right)\right)$; and (iii) discriminator $H_{d}$, the separate neural network that predicts domain label $\hat{d}=H_{d}\left(H_{f}\left(x\right)\right)$. In Figure 1, $H_{f}$ and $H_{p}$ correspond to the encoder and decoder parts of the model, respectively. $H_{d}$ is denoted with the dashed green arrow that passes the aggregated features to the adversarial loss.

Following [15], our optimization target is
(1)$$\begin{array}{c} \begin{array}{cc} E(\theta_{f},\theta_{p},\theta_{d}) & =\sum_{i=1}^{N_{s}}L_{s}\left(H_{p}\left(H_{f}\left(x_{i};\;\theta_{f}\right);\;\theta_{p}\right),\;y_{i}\right)\\ & -\lambda \sum_{j=1}^{N_{2}}L_{d}\left(H_{d}\left(H_{f}\left(x_{j};\;\theta_{f}\right);\;\theta_{d}\right),\;d_{j}\right)=\\ \; & =\sum_{i=1}^{N_{s}}L_{s}\left({\hat{y}}_{i},\;y_{i}\right)\;-\;\lambda \sum_{j=1}^{N_{2}}L_{d}\left({\hat{d}}_{j},\;d_{j}\right) \end{array} \end{array}$$
(2)$$\begin{array}{cc} \left({\hat{\theta }}_{f},{\hat{\theta }}_{p}\right) & =\underset{\theta_{f},\theta_{p}}{arg min}E\left(\theta_{f},\theta_{p},\theta_{d}\right), \end{array}$$
(3)$$\begin{array}{cc} {\hat{\theta }}_{d} & =\underset{\theta_{d}}{arg max}E\left(\theta_{f},\theta_{p},\theta_{d}\right), \end{array}$$
where $L_{s}$ is the segmentation loss (BCE), $L_{d}$ is the domain classification loss (BCE). $\theta_{f},\;\theta_{p},\;\theta_{d}$ are the parameters of $H_{f},\;H_{p},\;H_{d}$, respectively, and ${\hat{\theta }}_{f},\;{\hat{\theta }}_{p},\;{\hat{\theta }}_{d}$ are the solutions we seek. Parameter λ regulates the trade-off between the $L_{s}$ and $L_{d}$ objectives.

However, there is no consensus in the literature on how to connect the discriminator to a segmentation network [30]. We follow the findings [30] that the earlier U-Net layers contain more domain-specific information than the later ones and use features from the encoder layers as input to the discriminator. Our preliminary experiments also confirm that using encoder features is a better strategy. We detail this approach in Figure 1(6).

We also modify the architectural design of DANN for the segmentation task. The goal of the discriminator is to classify a kernel that is used to reconstruct the image. To aggregate features before the discriminator, we use 1×1 convolutions and interpolation to equalize the number of channels and spatial size; then, we concatenate the result. The discriminator consists of a sequence of fully-convolution layers followed by several fully-connected layers. We also use Leaky ReLU [31] and average pooling to avoid sparse gradients.

### 2.4. Cross-Domain Feature Maps Consistency

Similar to the adversarial approach, we propose to remove style-specific kernel information from the feature maps. However, we additionally exploit the paired nature of the unlabeled dataset $S_{2}$. Instead of the adversarial loss, we minimize the distance between feature maps of paired images. We use the same notations $H_{f}$, $H_{p}$, $\theta_{f}$, $\theta_{p}$, and $\hat{y}=H_{p}\left(H_{f}\left(x;\;\theta_{f}\right);\;\theta_{p}\right)$ as in Section 2.3. Further, we denote the feature vector for every image *x* as *f*, $f=H_{f}\left(x;\;\theta_{f}\right)$. For the paired image $\tilde{x}$, we use the similar notation $\tilde{f}$.

During the training, we minimize the sum of segmentation loss and distance between paired features (*f* and $\tilde{f}$). Thus, the optimization problem is
(4)$$\begin{array}{c} \begin{array}{cc} E(\theta_{f},\theta_{p}) & =\sum_{i=1}^{N_{s}}L_{s}\left(H_{p}\left(H_{f}\left(x_{i};\;\theta_{f}\right);\;\theta_{p}\right),\;y_{i}\right)\;\\ \; & +\;\alpha \sum_{j=1}^{N_{2}}L_{c}\left(H_{f}\left(x_{j};\theta_{f}\right),\;H_{f}\left({\tilde{x}}_{j};\theta_{f}\right)\right)\\ \; & =\sum_{i=1}^{N_{s}}L_{s}\left({\hat{y}}_{i},\;y_{i}\right)\;+\;\alpha \sum_{j=1}^{N_{2}}L_{c}\left(f_{j},\;{\tilde{f}}_{j}\right), \end{array} \end{array}$$
(5)$$\begin{array}{cc} \left({\hat{\theta }}_{f},{\hat{\theta }}_{p}\right) & =\underset{\theta_{f},\theta_{p}}{arg min}E\left(\theta_{f},\theta_{p}\right), \end{array}$$
where $L_{s}$ is the segmentation loss (BCE) and $L_{c}$ is the *consistency* loss. For the consistency loss, we use mean squared error (MSE) between paired feature maps. Parameter α regulates the trade-off between two objectives. We call this method *F-Consistency* since it enforces the consistency between paired feature maps.

Along with DANN, we present our method schematically in Figure 1(5). Note that we do not need any additional model, e.g., discriminator $H_{d}$, in the case of F-Consistency. The features are aggregated from the same encoder layers as in DANN.

### 2.5. Cross-Domain Predictions Consistency

A special case of F-Consistency is enforcing the consistency of paired predictions as the predictions are de facto the feature maps of the last network layer. This approach is proposed in [32] also in the context of medical image segmentation. Further, we denote this method *P-Consistency*. Visually, it could be compared with DANN and F-Consistency in Figure 1(4). The optimization problem is the same as in Equation (Equation 4), except $L_{c}$ is Dice Loss [33] and *f* and $\tilde{f}$ are the last layer features, i.e., predictions.

## 3. Data

In our experiments, we use a combination of different datasets with chest CT images. The data could be divided into two subsets according to the experimental purposes. The first collection of datasets consists of images with annotated COVID-19 lesions, i.e., with binary masks of ground-glass opacity and consolidation. It serves to train the COVID-19 segmentation algorithms. We detail every dataset of the segmentation collection in Section 3.1.

The second collection consists of chest CT images which are reconstructed with different kernels but have no COVID-19 annotations. We filter the data into pairs of images reconstructed with the smooth and sharp kernels. This data is further used to adapt the models in an unsupervised manner. We detail the second collection in Section 3.2.

Most of the datasets are publicly available, making our experiments reproducible.

### 3.1. Segmentation Data

We use three publicly available datasets to train and test the segmentation algorithms: Mosmed-1110 [34], MIDRC [9], and Medseg-9. We ensure that selected datasets contain original 3D chest CT imaging studies without third-party preprocessing artifacts. The images from the Mosmed-1110 and MIDRC datasets are reconstructed using smooth kernels, whereas Medseg-9 images have sharp reconstruction kernels. That allows us firstly to identify and then address the domain adaptation setup. We split the segmentation data into the source (COVID-train) and target (COVID-test) domains and describe them in Section 3.1.1 and Section 3.1.2, respectively. A summary of the segmentation datasets is presented in Table 1.

#### 3.1.1. COVID-Train

##### Mosmed-1110

This dataset consists of 1110 chest CT scans collected in Moscow clinics during the first months of 2020 [34]. Scanning was performed on *Canon (Toshiba) Aquilion 64* units using the standard scanner’s protocol: inter-slice distance of 0.8 mm and smooth reconstruction kernels in particular. However, the public version of Mosmed-1110 contains every 10th slice of the original series, which makes the resulting slice distance equal to 8.0 mm. Additionally, the 50 series have annotated binary masks depicting COVID-19 lesions (ground-glass opacity and consolidation). We use only these 50 images in our experiments.

##### MIDRC

MIDRC-RICORD-1a is the publicly available dataset that contains 120 chest CT studies [9]. The total number of volumetric series is 154. According to the DICOM entries, all images have smooth reconstruction kernels. The dataset contains at least 12 paired images (without considering the studies that contain more than two series). However, we do not use these pairs to enforce consistency since both images have smooth kernels. Also, we use only the images that have non-empty annotations. The images that have empty binary masks of COVID-19 are discarded from both of the datasets. The resulting training dataset consists of 112 volumetric images with smooth kernels.

#### 3.1.2. COVID-Test

##### Medseg-9

MedSeg website (https://medicalsegmentation.com/covid19/ (accessed on 24 August 2022)) shares a publicly available dataset with 9 annotated chest CT images from Radiopaedia.org (https://radiopaedia.org/articles/covid-19-3 (accessed on 24 August 2022)), reconstructed with sharp kernels. Contrary to the COVID-train dataset, Medseg-9 contains annotated lung masks. However, we find the masks predicted by our algorithm (see Section 4.1) more precise and use the predicted ones.

### 3.2. Paired Images Data

To train and evaluate the consistency of the segmentation algorithms, we use two sources of paired data. The first source is a publicly available dataset Cancer-500 [16]. But the Cancer-500 dataset does not contain COVID-19 cases. Thus, to properly evaluate the consistency of COVID-19 Cancer-500segmentation algorithms, we use the second source of private data that contains COVID-19 cases.

From Cancer-500, we build the Paired-public dataset (Section 3.2.1) and use it only to train the segmentation algorithms in an unsupervised manner. Then, we build the Paired-private dataset from our private data (Section 3.2.2). Besides training, we use this dataset to evaluate the consistency scores since it contains the segmentation target (COVID-19 lesions). The summary of the paired datasets is presented in Table 2.

Both datasets do not contain any COVID-19 or lung annotations. Note that we do not need COVID-19 annotations since we use these datasets in unsupervised training. However, we need lung masks to preprocess images. Thus, we use the same standalone lungs segmentation model as for the other datasets.

#### 3.2.1. Paired-Public

We build the Paired-public dataset using a publicly available dataset, Cancer-500 [16]. The data was collected from 536 randomly selected patients of Moscow clinics in 2018. All original images were obtained using a *Toshiba* scanner and reconstructed with FC07, FC51, or FC55 kernels. Here, FC07 is a smooth reconstruction kernel, whereas FC51 and FC55 are sharp kernels. From 536 studies, we extracted 120 pairs comparing their acquisition time and slice locations of the corresponding DICOM series. The latter two-step procedure leaves us with the image pairs that have voxel-to-voxel anatomical correspondence. We also filter contrast-enhanced cases. As a result, the Paired-public dataset consists of 98 FC07/FC51 and 22 FC07/FC55 pairs (Table 2). We use this dataset to train the domain adaptation algorithms on paired images.

However, the Paired-public dataset does not contain COVID-19 cases (it was collected before the pandemic). The latter observation limits using this dataset to evaluate the consistency. Otherwise, we evaluate the quality of COVID-19 segmentation algorithms using images with no COVID-19 lesions. Thus, we either evaluate the consistency of noisy or false positive predictions. For the same reason, one should also be careful using this data to enforce consistency in the last network layers, e.g., in P-Consistency (Section 2.5). The data without COVID-19 lesions can force the network to output trivial predictions.

Therefore, we introduce a private dataset for the extended consistency evaluation.

#### 3.2.2. Paired-Private

From the private collection of the chest CT images, we filter out 181 pairs to create the Paired-private dataset. These images were initially collected from Moscow outpatient clinics during the year 2020. Scanning was performed on the *Toshiba* and *GE medical systems* units using diverse settings. We select the four most frequent kernel pairs with a total of six unique reconstruction kernels: FC07, FC51, FC55 from *Toshiba*, and STANDARD, LUNG, SOFT from *GE medical systems* scanners. We detail the distribution of kernel pairs in Table 2.

Due to the purpose of collecting the data, these images contain COVID-19 lesions. Therefore, we use the Paired-private dataset both for training and evaluation. The wider variety of kernels also allows us to test the generalization of algorithms to unseen kernels. More specifically, we consider kernels LUNG, SOFT, and STANDARD in the Paired-private dataset to be unseen for the model trained on Paired-public data or only FC07, FC51, and FC55 kernels (excluding the rest) of the Paired-private dataset.

## 4. Experiments

The main focus of the experiments below is to compare our method to the other unsupervised domain adaptation techniques. To achieve an objective comparison, a fair and unified experimental environment should be created. Therefore, we firstly describe the common preliminary steps that build up every method. This description includes preprocessing and training details in Section 4.1.

Further, we detail every of the considered domain adaptation methods: Filtered Backprojection Augmentation (FBPAug) in Section 4.2, Domain Adversarial Neural Network (DANN) in Section 4.3, cross-domain feature-maps consistency (F-Consistency) which is our method in Section 4.4, and cross-domain predictions consistency (P-Consistency) in Section 4.5.

In all experiments, we use the same data split and evaluation metrics. Firstly, we split the COVID-train dataset (source domain with smooth kernels) into 5 folds. Then, we perform a standard cross-validation procedure, training on the data from four folds and calculating the score on the remaining fold. Here, we calculate the Dice Score between the predicted and ground truth COVID-19 masks for every 3D image and average these scores for the whole fold. Also, for every validation, we calculate the average Dice Score on the COVID-test dataset, which is the target domain with sharp kernels. We report the mean and standard deviation of these five scores on cross-validation and target domain data.

Besides the Dice Score on the source and target data, we also report the Dice Score between predictions on the paired images. To do so, we split the Paired-private datasets’ pairs into training and testing folds stratified by the type of kernel pairs. The size of the test fold is approximately 30% of the dataset size. Then, we supplement the source domain training data (four current folds of the cross-validation) with the fixed training part of the paired data. Average Dice Score is calculated on the test part in a similar fashion.

### 4.1. COVID-19 Segmentation

#### 4.1.1. Preprocessing

Before passing to the segmentation model, we apply the same preprocessing pipeline to all CT images. Preprocessing consists of four steps. (i) We rescale a CT image to have 1.75×1.75 mm axial resolution using linear interpolation. (ii) Then, the intensity values are clipped to the minimum of −1000 Hounsfield units (HU) and a maximum of 300 HU. (iii) The resulting intensities are min-max-scaled to the [0;1] range. (iv) Finally, we crop the resulting image to the automatically generated lung mask.

We generate the lung mask by training a standalone CNN segmentation model. The training procedure and architecture are reproduced from [8]. The training of the lung segmentation model involves two external chest CT datasets: LUNA16 [35,36] and NSCLC-Radiomics [37,38]. These datasets have no intersection with the other datasets used to train the COVID-19 segmentation models; thus, there is no leak of the test data.

The trained lung segmentation model achieves about 0.98 Dice Score both on the cross-validation and on the Medseg-9 dataset (it contains annotated lung masks). The latter result indicates that the lung segmentation model is robust to different kinds of domain shift and, moreover to the appearance of novel lesions (COVID-19).

#### 4.1.2. Training Details

In all COVID-19 segmentation experiments, we use the same 2D U-Net architecture described in Section 2.1. We train all models for 25k iterations using Adam [39] optimizer with the default parameters and an initial learning rate of $10^{-4}$. Every 6k batches learning rate is multiplied by 0.2. Each iteration contains 32 randomly sampled 2D axial slices. Training of the plain segmentation model and similarly F-Consistency takes approximately 8 h on Nvidia GTX 1080 (8 GB) (Santa Clara, CA, USA). Training of DANN and FBPAug takes 12 and 30 h, respectively, in the same setting. The inference time is approximately 5–10 seconds depending on the image size and it is the same for all methods.

We further call the model trained only on the source data without any adaptation a **baseline**. We refer to the baseline scores on the COVID-test dataset and its consistency scores on Paired-private as a starting point for all other methods.

### 4.2. Filtered Backprojection Augmentation

The first method that we consider is FBPAug [14]. In our experiments, we use the original implementation of FBPAug from [14]. However, we sample parameters from the interval that corresponds to only *sharp* reconstruction kernels (*a* from [10.0,40.0], *b* from [1.0,4.0]), as our goal is to adapt the model to *sharp* kernels. We use the same notations of *a* and *b* as in [14]. We also reduce the probability of augmenting an image from 0.5 to 0.1. The latter change does not affect the performance (tested on a single validation fold) and reduces the experiment time (saving about 90 h per experiment).

Note that the experimental setup remains the same as in baseline (Section 4.1). FBPAug is the only adaptation method that does not use paired data.

### 4.3. Domain Adversarial Neural Network

The next step is to use unlabeled paired data to build a robust domain shift algorithm. Here, we adopt a DANN approach [15] to the COVID-19 segmentation task.

In our experiments, we use the scheduling of parameter λ as in [15]. The baseline training procedure is extended to samples from the unlabeled data. At every iteration, we additionally sample 16 pairs of axial slices (the batch size is 32) from one of the Paired-private or Paired-public datasets (depending on a data setup). Then, we make the second forward pass to the discriminator and sum the segmentation and adversarial losses. The rest of the pipeline remains the same as in the baseline.

For this method, we select two parameters that can drastically change its behavior in terms of consistency and segmentation quality. Firstly, we evaluate different λ values, where λ determines how strongly adversarial loss contributes to the total loss. With the close to zero λ, we expect DANN to behave similarly to baseline. With the larger λ, we expect our segmentation model to fool the discriminator, making features of the different kernels indistinguishable for the latter. However, this consequence does not guarantee an increase in consistency or segmentation quality. Therefore, we manually search for the $\lambda \in \{10^{-5},10^{-4},10^{-3},10^{-2},10^{-1},1\}$ and choose the best by the cross-validation score.

Finally, we evaluate how well DANN generalizes under the presence of different kernel pairs. To do so, we exclude SOFT/LUNG and STANDARD/LUNG kernel pairs from training. We compare the results of this experiment with the model that is trained on all available kernel pairs from Paired-private. We also test the sensitivity of the DANN approach to the presence of COVID-19 lesions in the unlabeled data. In this case, we train DANN on the Paired-public dataset that does not contain COVID-19 targets.

### 4.4. Cross-Domain Feature Maps Consistency

Our proposed F-Consistency also uses unlabeled paired data. Therefore, the training procedure is the same as for DANN (Section 4.3), except we do not use any scheduling for parameter α.

Similarly to DANN’s experimental setup, we evaluate different α values and choose the best. Here, α controls the trade-off between consistency and segmentation quality. However, contrary to the discriminator’s λ in Section 4.3, the large α values for consistency regularization ensure the alignment of the features. We show this trade-off for ten α values in a log-space from $3^{-10}$ to 1.

Finally, we evaluate the generalization of F-Consistency to different kernel pairs similarly by excluding SOFT/LUNG and STANDARD/LUNG kernel pairs from training. Similar to DANN, we train our method on the Paired-public dataset that does not contain COVID-19 lesions and show its generalization, regardless of the semantic content.

### 4.5. Cross-Domain Predictions Consistency

One special case of F-Consistency is enforcing the paired predictions consistency, which is independently evaluated in [32]. We call this case a P-Consistency. We follow the experimental setup as in F-Consistency (Section 4.4) and show the trade-off between the target and consistency scores for the same values of α.

For the P-Consistency, we draw one’s attention to the experiment on the Paired-public dataset. Since this dataset does not contain COVID-19 cases, the consistency of the enforced predictions on empty-target images can result in trivial predictions. Thus, we expect P-Consistency to be less generalizable to the target domain in terms of target Dice Score.

## 5. Results

We structure our experimental results as follows. We firstly compare all methods in Section 5.1 so we directly support our main message. Secondly, we compare the generalization of all methods trained on fewer data in Section 5.2. Then, we visualize the trade-off between the consistency and COVID-19 segmentation quality in Section 5.3.

We compare some of the key results statistically using a one-sided Wilcoxon signed-rank test. We report *p*-values in two testing setups: $p_{1}$, comparing five mean values after cross-validation, and $p_{2}$, comparing Dice Score on every image as an independent sample.

### 5.1. Methods Comparison

To begin with, we show the existence of the domain shift problem within the COVID-19 segmentation task. The Dice Score of the baseline model on the COVID-test dataset is lower than the cross-validation score on the COVID-train dataset, 0.56 against 0.60. Also, this score is significantly lower than 0.64 achieved by our adaptation method $\left(p_{1}<0.05,\;p_{2}<10^{-4}\right)$. Results could be found in Table 3 comparing row Baseline to the others. For all adaptation methods, we observe an increase in the consistency score and segmentation quality on the target domain. Moreover, all methods maintain their quality on the source domain compared to Baseline.

Further, we compare FBPAug to the best adaptation methods since it is a straightforward solution to the domain shift problem caused by the difference in the reconstruction kernels (Section 2.2). Although FBPAug achieves comparable results on the target domain, our method outperforms it in terms of the average consistency score, 0.80 Dice Score against 0.76
$\left(p_{1}<0.05,\;p_{2}<10^{-5}\right)$. The results are also in Table 3, row FBPAug.

Then, we compare P-Consistency (which operates with the last layer of decoder) with F-Consistency and show it resulting in the significantly lower consistency score, 0.63 against 0.80
$\left(p_{1}<0.05,\;p_{2}<10^{-10}\right)$. Thus, our findings align with the message [30] that the encoder layers contain more domain-specific information than the decoder ones.

F-consistency outperforms DANN. Although both methods score similar in target Dice Score, F-Consistency has an advantage over DANN in the consistency score: 0.80 against 0.78
$\left(p_{1}<0.05,\;p_{2}<10^{-2}\right)$. Our intuition here is F-Consistency explicitly enforces features alignment, while DANN enforces features to be indistinguishable for the discriminator. The latter differently impact the consistency score, and F-Consistency performs better.

We conclude the comparison of the methods with the qualitative analysis. In Figure 2, one could find examples of the Baseline, FBPAug, DANN, and F-Consistency predictions on the COVID-test dataset and compare them with the ground truth. Although all adaptation methods perform similar to the ground truth with minor inaccuracies, Baseline outputs the massive false positive predictions on the unseen domain. Additionally to the quantitative analysis above, the latter observation highlights the relevance of the domain adaptation problem in the COVID-19 segmentation task.

In Figure 3, we visualize predictions of the same four methods on the paired images from the Paired-private dataset. For the Baseline, we observe an extreme inconsistency (Figure 3A) and massive false positive predictions in healthy lung tissues (Figure 3D) and even outside lungs (Figure 3B). For the adaptation methods, their predictions are visually more consistent inside every pair, which aligns with the consistency scores in Table 3. Despite the high consistency scores, FBPAug and DANN output more aggressive predictions. FBPAug predicts motion artifacts near the body regions (Figure 3A) and triggers similarly to the baseline on, most likely, healthy lung tissues (Figure 3B). DANN is more conservative but triggers on the consolidation-like tissues (Figure 3C,D). However, without the ground truth annotations on the paired data, we refer to this analysis as a discussion.

Below, we investigate the generalization of the models trained with fewer data and the trade-off between consistency and segmentation quality.

### 5.2. Generalization with the Less Data

Firstly, we show how DANN, P-Consistency, and F-Consistency generalize to the unseen reconstruction kernels. We remove SOFT/LUNG and STANDARD/LUNG pairs of the Paired-private dataset from training, so we train the models using FC07/FC51 and FC07/FC55 pairs. The results of the removed kernel pairs are shown in Table 4.

The methods preserve their segmentation quality on the COVID-train and COVID-test datasets despite training them with limited data. Moreover, all three methods score considerably higher than Baseline in consistency scores for unseen kernel pairs (SOFT/LUNG and STANDARD/LUNG). The latter means that the adaptation methods manage to align stylistic-related features even from the limited number of training examples. However, we highlight a decrease in the average consistency scores compared to the versions trained on full Paired-private. At this point, FBPAug (Table 3) outperforms the adaptation methods. The latter indicates that the range of synthetically augmented data overlaps the range of reduced Paired-private.

Further, we evaluate the models regularized using paired images from the Paired-public dataset. The dataset contains only FC07/FC51 and FC07/FC55 kernel pairs. Besides the previous setup, this data does not contain COVID-19 lesions. Thus, we demonstrate that some methods depend on the semantic content and poorly generalize to kernel styles. The results are shown in Table 5.

We highlight two main findings from these results. Firstly, consistency of the method that operates with the decoder layers decreases to the level of Baseline; see P-Consistency in Table 5. Our intuition here is that the decoder version of the model can be more easily enforced to output the trivial predictions than the encoder one. Simultaneously, the images without COVID-19 lesions induce trivial predictions. Therefore, it might be easier for these models to differ the paired dataset from the source dataset by the semantic content and fail to align the stylistic features. Finally, we observe our method, F-Consistency, to outperform the other adaptation methods training only on the publicly available data.

### 5.3. Trade-Off between Consistency and Segmentation Quality

The main problem with maximizing the consistency score is converging to the trivial solution (empty predictions). We vary α, the parameter that balances the consistency and segmentation losses, for P-Consistency and F-Consistency; see Figure 4. The resulting trade-off follows the expected trend: consistency increases to 1 and target Dice Score decreases to 0 with increasing α.

We use Dice Score on the COVID-train dataset as a perceptual criterion to choose α. We stop at the largest α value before Dice Score starts to drop: $10^{-3}$ for P-Consistency and 1 for F-Consistency. However, we use Paired-private, which participates in the final comparison, to calculate the consistency score. Firstly, we argue that using the Paired-public dataset in this setup is incorrect. Paired-public does not contain COVID-19 lesions; thus, we can only measure the consistency of false positive predictions. Secondly, we choose α without considering consistency scores. Thus, we also do not overfit under the consistency scores.

For the DANN method, we choose $\lambda =10^{-2}$ based on the best score on COVID-train. Far from the optimal λ values, DANN’s prediction scores have a large standard deviation, so the trade-off cannot be observed. We also note that the adversarial approach does not explicitly enforce trivial predictions. Hence, we report the trade-off only for the F-Consistency and P-Consistency methods.

## 6. Discussion

Here, we summarize our results, discuss the most important limitations of our study, and suggest possible directions for future work.

We have shown that the proposed F-consistency significantly improves the performance on the target domain compared to the baseline model. However, we do not train the oracle model, which indicates the upper bound for other methods in a domain adaptation task. The oracle model should be trained via cross-validation on the target domain. In our case, the target domain contains only nine images, which leads either to lower results due to the small size of the training set or high dispersion of the results. Therefore, we compare the adaptation methods only with the baseline model and between each other.

In Section 5.1, We show our model to achieve the highest results in terms of the consistency score. Contrary, the authors of [32] observing a tendency of models to converge to trivial solutions using consistency loss. They assume that the models learn to distinguish domains for which they are penalized; thus, the models yield trivial but perfectly consistent predictions. Although we run the same setup with [32], we do not observe trivial predictions for our method. The latter is demonstrated through the whole Section 5. Our intuition here is that the inner structure of domains and the semantic content of images in our case are more diverse, preventing the model from overfitting under a specific domain.

We highlight that adversarial and consistency-based methods depend on a diverse unlabeled pool of data; see Section 5.2. On the other hand, FBPAug does not require additional data since it augments the images from source dataset. One could think of this method as enforcing the consistency between the original and augmented image predictions using ground truth as a proxy. Simultaneously, we show that the models operating with the earlier layers perform better. Therefore, we could train F-Consistency on the pairs of original and augmented with FBPAug images to achieve even better results. We leave the latter idea for future work.

### Conclusions

We have proposed an unsupervised domain adaptation method, F-Consistency, to address the difference in CT reconstruction kernels. Our method uses a set of unlabeled CT image pairs and enforces the similarity between feature maps of paired images. We have shown that F-Consistency outperformed the other adaptation and augmentation approaches in the COVID-19 segmentation task when provided with enough unlabeled data. Finally, through extensive evaluation, we have shown our method to generalize the unseen reconstruction kernels better and without the specific semantic content.

## Figures and Tables

**Figure 1 jimaging-08-00234-f001:**
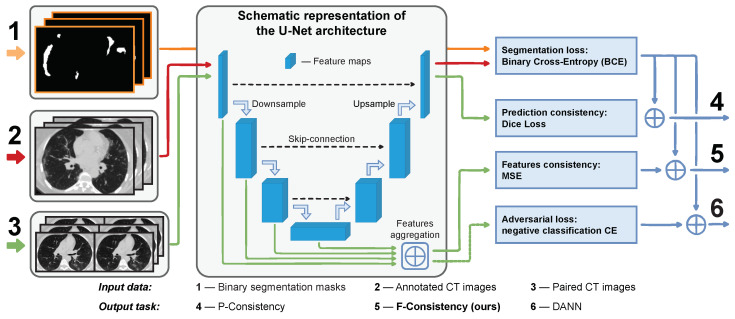
Schematic representation of the proposed method, ***F-Consistency*** (**5**), and its competitors, *P-Consistency* (**4**) and *DANN* (**6**). All methods build upon the same U-Net architecture, which we train to segment the COVID-19 binary mask (**1**) from the axial slices of chest CT images (**2**). These adaptation methods use unlabeled paired data (**3**) to improve the model performance on the target domain. We show the flow and different usage of the paired data in different methods with green.

**Figure 2 jimaging-08-00234-f002:**
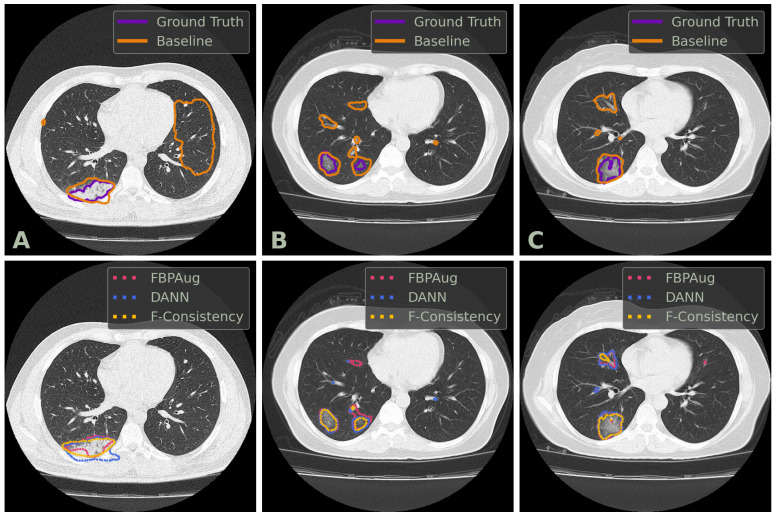
Examples of axial CT slices from the COVID-test dataset with the corresponding predictions and ground truth annotations. Three columns, denoted (**A**–**C**), contains three unique slices. The top row contains the contours of the ground truth and baseline prediction. The bottom row contains the contours of the adaptation methods’ predictions. DANN and F-Consistency correspond to DANN and F-Cons from Table 3, respectively.

**Figure 3 jimaging-08-00234-f003:**
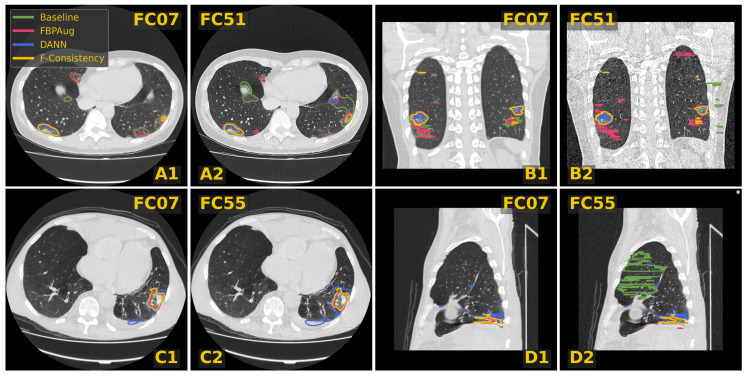
Examples of CT slices from the Private-paired dataset with the corresponding predictions on the paired images. Four doublets, denoted (**A**–**D**), contain corresponding slices from the smooth and sharp images. The doublets B and D are coronal and sagittal slices, respectively. Every slice contains predictions of four methods named in the legend.

**Figure 4 jimaging-08-00234-f004:**
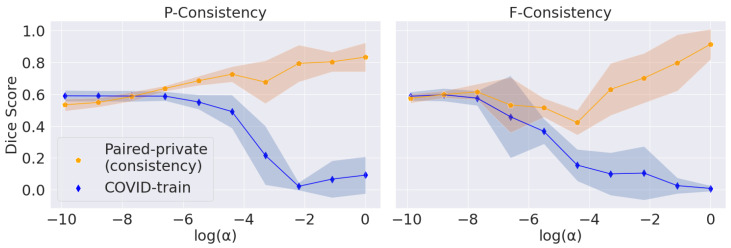
Trade-off between the segmentation quality and consistency scores induced by the regularization parameter α (Section 2.4). The blue line corresponds to Dice Scores calculated on the COVID-train dataset. The orange line corresponds to the consistency scores calculated on the Paired-private dataset. The shaded areas correspond to the standard deviation along the Y-axis.

**Table 1 jimaging-08-00234-t001:** Summary of the segmentation datasets. *Effective size* means the number of annotated images after appropriate filtering.

Dataset	Source	Effective Size	Kernels	Annotations	Split
COVID-train	Mosmed-1110 [34]	50	unknown smooth	COVID-19 mask	5-fold cross-val
	MIDRC [9]	112	B/L/BONE/ STANDARD (smooth)	COVID-19 mask	
COVID-test	Medseg-9	9	unknown sharp	COVID-19 mask, lungs mask	hold-out test

**Table 2 jimaging-08-00234-t002:** Summary of the datasets with paired images.

Dataset	Kernel Pair (Smooth/Sharp)	Training	Testing Pairs
Paired-public [16]	FC07/FC55	22	0
	FC07/FC51	98	0
Paired-private	FC07/FC55	60	20
	FC07/FC51	30	11
	SOFT/LUNG	30	10
	STANDARD/LUNG	30	10

**Table 3 jimaging-08-00234-t003:** Comparison of all considered methods. The adaptation methods are trained using all training kernel pairs of the Paired-private dataset. F-/P-Cons stand for F-/P-Consistency, where F-Consistency is our proposed method. All results are Dice Scores in the format *mean* ± *std* calculated from 5-fold cross-validation. We highlight the best scores in every column in **bold**.

	COVID-Train	COVID-Test	Paired-Private Consistency				
			FC07/55	FC07/51	SOFT/LUNG	STAND/LUNG	Mean
Baseline	0.60±0.04	0.56±0.03	0.52±0.06	0.39±0.07	0.58±0.08	0.28±0.05	0.46±0.05
FBPAug	0.59±0.04	0.62±0.03	0.80±0.02	0.71±0.03	0.85±0.01	0.65±0.03	0.76±0.02
DANN	0.58±0.05	0.64±0.02	0.84±0.02	0.70±0.02	0.86±0.03	0.66±0.06	0.78±0.02
P-Cons	0.59±0.04	0.61±0.01	0.65±0.05	0.60±0.02	0.77±0.01	0.47±0.04	0.63±0.03
F-Cons	0.57±0.03	0.64±0.03	0.88±0.01	0.72±0.04	0.83±0.02	0.70±0.05	0.80±0.01

**Table 4 jimaging-08-00234-t004:** Comparison of DANN, P-Consistency, and F-Consistency generalizing to previously unseen SOFT, STANDARD, and LUNG kernels. The numbers in the brackets next to the methods correspond to the number of kernel pairs in the Paired-private dataset they are trained with, e.g., DANN (4) matches with the DANN in Table 3. All results are Dice Scores in the format *mean* ± *std* calculated from 5-fold cross-validation.

	COVID-Train	COVID-Test	Paired-Private Consistency				
			FC07/55	FC07/51	SOFT/LUNG	STAND/LUNG	Mean
Baseline	0.60±0.04	0.56±0.03	0.52±0.06	0.39±0.07	0.58±0.08	0.28±0.05	0.46±0.05
FBPAug	0.59±0.04	0.62±0.03	0.80±0.02	0.71±0.03	0.85±0.01	0.65±0.03	0.76±0.02
DANN (4)	0.58±0.05	0.64±0.02	0.84±0.02	0.70±0.02	0.86±0.03	0.66±0.06	0.78±0.02
DANN (2)	0.59±0.05	0.64±0.02	0.81±0.03	0.70±0.03	0.74±0.02	0.58±0.07	0.73±0.02
P-Cons (4)	0.59±0.04	0.61±0.01	0.65±0.05	0.60±0.02	0.77±0.01	0.47±0.04	0.63±0.03
P-Cons (2)	0.59±0.04	0.59±0.03	0.62±0.03	0.56±0.02	0.72±0.01	0.40±0.04	0.59±0.02
F-Cons (4)	0.57±0.03	0.64±0.03	0.88±0.01	0.72±0.04	0.83±0.02	0.70±0.05	0.80±0.01
F-Cons (2)	0.58±0.04	0.64±0.01	0.83±0.02	0.64±0.03	0.75±0.02	0.59±0.02	0.73±0.01

**Table 5 jimaging-08-00234-t005:** Comparison of all adaptation methods from Table 3 except FBPAug trained on the Public-paired dataset. All results are Dice Scores in the format *mean* ± *std* calculated from 5-fold cross-validation. We highlight the consistency scores near or below Baseline level in *italic*. The best consistency scores are highlighted in **bold**.

	COVID-Train	COVID-Test	Paired-Private Consistency				
			FC07/55	FC07/51	LUNG/SOFT	LUNG/STAND	Mean
Baseline	0.60±0.04	0.56±0.03	0.52±0.06	0.39±0.07	0.58±0.08	0.28±0.05	0.46±0.05
DANN	0.60±0.03	0.64±0.02	0.75±0.02	0.64±0.05	0.67±0.03	0.50±0.05	0.66±0.02
P-Cons	0.53±0.03	0.58±0.03	0.54±0.05	0.44±0.03	0.57±0.04	0.28±0.06	0.47±0.03
F-Cons	0.59±0.04	0.64±0.02	0.80±0.02	0.63±0.04	0.71±0.02	0.55±0.05	0.70±0.02

## Data Availability

The chest CT scans from the Mosmed-1110, MIDRC-RICORD-1a, MedSeg-9, Cancer-500, LIDC/IDRI, and NSCLC-Radiomics datasets used in this study are publicly available. The chest CT scans from the Paired-private dataset used in this study are not publicly available for privacy reasons.

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
