# Peer review of "Adaptation to CT Reconstruction Kernels by Enforcing Cross-Domain Feature Maps Consistency"

_2313-433X, 2022, doi:10.3390/jimaging8090234_

Round 1

Reviewer 1 Report

In this paper, the authors propose a new method to address the issue of domain shift during segmentation of CT images in COVID-19. To address the root cause of domain shifting, which is the discrepancy in the reconstruction kernels, the authors propose an unsupervised adaptation method called F-Consistency that uses a set of unlabeled CT image pairs and enforces similarity between this image pair by minimizing the mean square error. The performance of this method is compared with other adaptation methods like filtered backprojection and Deep Adversarial Neural networks. Through various numerical examples, it was shown that the F-Consistency method outperformed the other methods. 

The paper has been very well-written, easy-to-follow, and addresses a very important yet challenging question of accurate segmentation in CT images to identify minute COVID-19 characteristics for better diagnosis and treatment. A detailed literature study has been provided and the novelty of the proposed work has been clearly presented in Section 1.2. The methods and datasets used have been clearly described or referenced to. One suggestion for improvement might be to compare computational times for the proposed method with other methods besides the Dice and p-scores. Else, this paper merits publication in its current form.

Reviewer 2 Report

Overall the article is interesting and well-written. Including feature-consistency between domains as part of the optimization is an interesting and, to my knowledge, novel approach to extracting meaningful features from images for segmentation.  The authors have clearly put significant effort into the manuscript and research.  The results do feel somewhat incremental though when compared to DANN, and I still struggle to see an important use case for these unsupervised learning approaches if paired data is available (please see major issue #1).

Most of my critiques are minor, however I have two more major critiques that I would appreciate a response to, or should be addressed in a revision to the paper.  Please note that neither of these critiques change my overall judgement on the quality of the work.

There are also many grammatical errors that the authors should work with the editors to resolve.  Generally speaking though, these do not inhibit a reader's understanding of the work.

==================================================

Major issues:

1. Choice of "baseline"

The authors have chosen as their baseline a network trained only on the smooth images from their datasets, which to me is less of a "baseline" and more of a "naive" approach to the task.  It is not a particularly meaningful comparison, beyond a simple illustration that encoder-decoder-based segmentation networks do not translate well from smooth kernels to sharp kernels, if they have never been trained on sharp-kernel input data.  This is also less of a valid comparison since all of the other methods, to some extent, are trained (unsupervised) on sharp data.

I feel that a more appropriate baseline for the study would be a network trained on both the smooth *and* sharp kernel images, which is then evaluated on smooth and sharp kernel images.  From my reading, it appears that that authors have gone to great lengths to identify perfectly paired image datasets (i.e. same sinogram, same FOV, same reconstruction geometry, same slice location, etc.) with the *only* difference being the reconstruction kernel utilized.  Therefore, although technically the sharp images are unlabeled, the ground truth markings/segmentations are just as valid on the sharp kernel images as they are on the smooth kernel images.  Therefore, prior to publication, I think this needs to be utlized as the baseline, rather the currently provided baseline.

I also think the authors should motivate why their F-Cons approach and DANN are important if paired data is available.  Typically if the data are perfectly paired (as appears to be the case in this study), then labels should be easily transferrable between the paired datasets.

2. Inclusion of the *P-Cons (dec)* and *F-cons (dec)* and *DANN (dec)*

I felt that overall the inclusion of the decoder-based consistency methods (as opposed to the encoder based approaches) did not add much value to the work overall, and generally made the work harder to read.  The work is already quite dense with lots of good results and information, and I think the authors should consider moving the comparision between (Dec)- and (Enc)-based approaches either to a separate article, or into an appendix.  Moreover, I did not feel that much motivation was given for *why* one might expect signficant differences in performance between the two.  All-in-all, unless better motivation can be provided, and clarity can be improved, I would suggest leaving this out of the primary analysis beyond perhaps a comment in the discussion.  Particularly since these results simply agree with/support the results of [31].

==================================================

Minor Issues:

Figure 3: Images should probably be larger.  Some of the points illustrated and the image details are not clear at their current sizes.

Line 111-112: Word choice here is confusing. I think the authors are trying to say "our domain adaptation method could be *translated* from segmentation *to* classification..."

Line 129: Grammatically incorrect. Remove "the" from "with the consolidation..."

Line 189-195: Not clear if we're discussion the default DANN implementation or the authors' modifications

Line 267: Gramatically incorrect. Remove "further".

Line 285: Change "here" (i.e. the reference to the link) to be the name of the dataset/website, similar to how the link for "Medseg Website" appears in line 284.  Use of "here" doesn't make sense in the article text.

Line 308: Please explain more clearly what you mean by "comparing the shape."  FOV? Center of the reconstruction volume? Etc.  Are the scans paired in such a way that there is a one-to-one correspondence between voxels?

Line 330: Please clarify that the difference between the FC* kernels and the LUNG, SOFT, STANDARD kernels is manufacturer specific.

Line 359-368:  The way this section is written is confusing.  Segmentation of the lungs here is technically a preprocessing step, but by dividing it into two separate paragraphs with headers, it makes the order of operations confusion.  I would consider putting it all under the heading of "preprocessing" and then discussing the details of the lung segmentation network (364-373) elsewhere in the paper (maybe an appendix)?  Please clarify the exact order of operations here for all preprocessing.

Line 378: "nVidia" capitalization is incorrect.  Please change to "Nvidia."

Line 456: "One could find the results..." is not very clear.  More clear would be "Results can be found..."

Line 473-481: This section is particularly challenging to read and make sense of.  I think it's a good reflection of major point 2 above, where the included analysis of the F-cons (dec) actually detracts from the other points the authors are making.  There is a fair bit of jargon, shortened names (although inconsistently shortened), and parenthetical p-values, that all make this section quite challenging to understand and extract the results the authors wish to make.  Please rework to improve clarity and readability.

Line 477: If you are going to use the shortened "F-cons" term, please also shorten "P-consistency" to be "P-cons."  (Although, I feel that overall this section may be more readable if the full names of F-consistency and P-consistency are used throughout).

Line 482: Gramatically incorrect.  More clear would be "F-consistency outperforms P-consistency and DANN.".

Line 513: Gramatically incorrect. Change "datasets despite we train them with limited data" to "datasets despite training them with limited data"

Line 527: Type: operates -> operate
